# Genetic and mechanistic basis for APOBEC3H alternative splicing, retrovirus restriction, and counteraction by HIV-1 protease

Diako Ebrahimi [1], Christopher M. Richards[1], Michael A. Carpenter[1,2], Jiayi Wang[1], Terumasa Ikeda[1,2], Jordan T. Becker[1], Adam Z. Cheng[1], Jennifer L. McCann[1], Nadine M. Shaban[1], Daniel J. Salamango[1], Gabriel J. Starrett[1,4], Jairam R. Lingappa[3], Jeongsik Yong[1], William L. Brown[1] & Reuben S. Harris[1,2]

Human APOBEC3H (A3H) is a single-stranded DNA cytosine deaminase that inhibits HIV-1. Seven haplotypes (I–VII) and four splice variants (SV154/182/183/200) with differing antiviral activities and geographic distributions have been described, but the genetic and mechanistic basis for variant expression and function remains unclear. Using a combined bioinformatic/experimental analysis, we find that SV200 expression is specific to haplotype II, which is primarily found in sub-Saharan Africa. The underlying genetic mechanism for differential mRNA splicing is an ancient intronic deletion [del(ctc)] within A3H haplotype II sequence. We show that SV200 is at least fourfold more HIV-1 restrictive than other A3H splice variants. To counteract this elevated antiviral activity, HIV-1 protease cleaves SV200 into a shorter, less restrictive isoform. Our analyses indicate that, in addition to Vif-mediated degradation, HIV-1 may use protease as a counter-defense mechanism against A3H in >80% of sub-Saharan African populations.

[1] Department of Biochemistry, Molecular Biology and Biophysics, Masonic Cancer Center, Institute for Molecular Virology, Center for Genome Engineering, University of Minnesota, Minneapolis, MN 55455, USA. [2] Howard Hughes Medical Institute, University of Minnesota, Minneapolis, MN 55455, USA. [3] Departments of Global Health, Medicine and Pediatrics, University of Washington, Seattle, WA 98104, USA. [4] Present address: Center for Cancer Research, National Cancer Institute, National Institutes of Health, Bethesda, MD 20892, USA. These authors contributed equally: Diako Ebrahimi, Christopher M. Richards. Correspondence and requests for materials should be addressed to R.S.H. (email: rsh@umn.edu)

Most humans have seven *APOBEC3* (apolipoprotein B mRNA editing enzyme, catalytic polypeptide-like 3, *A3*) genes organized in tandem on chromosome 22 and located between conserved chromobox genes *CBX6* and *CBX7*[1]. These genes, *A3A-D* and *A3F-H*, encode members of the APOBEC family of polynucleotide cytosine deaminases (reviewed in refs. [2,3]). The hallmark biochemical activity of this family is single-stranded DNA cytosine-to-uracil deamination, although the family namesake (APOBEC1) and at least one A3 enzyme (A3A) also have the capability to edit RNA cytosines[4,5]. These enzymes have diverse functions in biology ranging from transcriptome editing (APOBEC1 and A3A), antibody gene diversification (AID), and restriction of a broad number of DNA-based parasites (APOBEC1, AID, and A3A-H) (reviewed by refs. [2,3,6–8]). The principle function of A3 enzymes in innate immunity is evidenced by high degrees of genetic variation including large and small changes both between and within species. For instance, gene copy numbers vary from 1 gene in rodents (encoding 2 deaminase domains)[9], 7 genes in most humans (11 deaminase domains)[9,10], and up to 13 genes in bats (18 deaminase domains)[11]. Many humans lack the entire *A3B* coding sequence due to a 26.5 kb deletion[12] and/or *A3H* function due to a 3 bp deletion removing an essential amino acid[13]. Additionally, there are coding and non-coding single-nucleotide polymorphisms (SNPs) and insertion/deletion polymorphisms (indels), which cumulatively account for high rates of positive selection (reviewed by refs. [14–16]).

The best-studied A3 substrate is the retrovirus HIV-1 (reviewed by refs. [6,7,17]). Four different A3 enzymes, A3D, A3F, A3G, and A3H, have the capacity to package into assembling viral particles, track within particles until a new host cell becomes infected, and then target virus replication intermediates, e.g., [18–20]. Each A3 enzyme can interfere with the overall reverse transcription process to varying degrees by at least two distinct mechanisms. The first is a deamination-independent process by which A3 enzymes bind viral genomic RNA and directly impede reverse transcription, e.g., [21–23]. The second is a deamination-dependent process in which viral cDNA cytosines are converted to uracils, which become immortalized by reverse transcription into genomic strand G-to-A mutations, e.g., [24–26]. Left unchecked, mono or combinatorial A3 restriction activity can fully suppress virus replication, e.g., [27–29]. Therefore, it is not surprising that HIV-1 and related lentiviruses have an A3 counter-defense mechanism governed by the viral protein Vif, which nucleates the formation of an E3 ubiquitin ligase complex to degrade cellular A3 enzymes and prevent access to susceptible viral replication intermediates (reviewed by refs. [6,7,17]). Thus, during every round of virus replication, there is a continuous interaction between A3 enzymes attempting to restrict virus replication and Vif working to neutralize the A3 enzymes and preserve viral genomic integrity. The balance between restriction and counteraction is most likely to be sensitive to perturbation as proviral sequences from patients often have G-to-A mutations, e.g., [30,31], and A3 mutagenesis has been linked to virus evolution including immune escape and drug resistance[32–36].

*A3H* is the most variable *A3* gene in the human population. Four SNPs and one indel have been reported within *A3H* coding regions: N15/ΔN15, R18/L18, G105/R105, K121/D121, and E178/D178[13,37]. These natural variations have been organized non-randomly into seven different haplotype blocks (I–VII)[13,38] and two in particular have a major impact on A3H function. The deletion of N15 common to haplotypes III, IV, and VI results in a non-functional protein, likely due to loss of structural integrity and rapid degradation (i.e., undetectable by immunoblots and activity assays)[13,19,23,38]. G105 occurs in haplotypes I and VI (although inconsequential in the latter due to ΔN15) and

compromises antiviral activity in large part due to poor expression and altered subcellular localization[13,19,38–41]. Only haplotypes II, V, and VII have demonstrated potent restriction activity against Vif-deficient HIV-1 in model systems[19,20,38,39,42] and, due to a high global allele frequency and strong geographic biases, only *A3H* haplotype II is likely to contribute to HIV-1 restriction and mutagenesis in vivo[19,20,43,44]. For instance, although the global frequency of the *A3H* haplotype II is 20.9%, the allele frequency in HIV-1 pandemic areas of sub-Saharan Africa is 57%, which means 82% of this population has at least one allele of this HIV-1 restrictive haplotype.

*A3H* is further diversified by alternative splicing, with four reported splice variants (SV154, SV182, SV183, SV200)[39]. SV154 is predicted to be corrupted structurally and non-functional, SV182 and SV183 are formed by utilization of tandem alternative splice acceptors and have been shown to be similar functionally in prior experiments, and SV200 has a C-terminal extension that could potentially influence multiple activities relevant to HIV-1 restriction[23,39]. Here we endeavor to connect genotype to phenotype by asking whether a SNP (or multiple SNPs) within the *A3H* gene is responsible for alternative splicing and production of SV200. Genetic association and reporter gene studies combined to demonstrate that a single non-coding variation, a 3 nucleotide deletion in intron 4 (Δctc), is embedded within the haplotype II block and responsible for SV200 mRNA expression. Immunoblots of lymphoblastoid cell lines (LCLs) and peripheral blood mononuclear cells (PBMCs) showed that the SV200 enzyme is expressed in cells with haplotype II genotypes. HIV-1 infectivity experiments showed that the A3H haplotype II SV200 enzyme is at least fourfold more restrictive than other splice variants and, interestingly, that this enhanced activity is counteracted by a single cleavage event catalyzed by HIV-1 protease inside virus particles. Thus, our studies have uncovered a novel protease-dependent mechanism used by HIV-1, in addition to Vif, to counteract A3H-mediated restriction. This cleavage mechanism may be required by transmitting viral isolates in pandemic regions such as sub-Saharan Africa to counteract the increased restriction potency of cells with haplotype II genotypes.

## Results

**A3H genetic variation in humans.** Human *A3H* is polymorphic at five codons, and these variants are organized non-randomly into seven different haplotype blocks[13,38] (Fig. 1a, b and Table 1). To search for additional haplotypes, we analyzed the *A3H* genotypes of 2504 individuals from the 1000 Genomes Project[45]. These analyses confirmed six of the previously described *A3H* haplotypes (not haplotype VII), and also revealed the existence of six additional minor haplotypes, VIII through XIII. It is noteworthy that three of these new haplotypes have allele frequencies that exceed those of three previously described haplotypes (Table 1). Thus, humans collectively express at least 12 different *A3H* haplotypes out of the maximum number of 32 combinations (i.e., tight genetic linkage prevents random assortment by recombination of the two different alleles for each of the five variable codons).

**A3H haplotypes I and II are spliced differentially.** Our analyses of 1000 Genomes Project data confirmed the prevalence of haplotypes I, II, III, and IV in human populations[13,38] (i.e., allele frequencies > 1%; Table 1 and Fig. 2a). *A3H* haplotypes III and IV are not detectable at the protein level, even upon exogenous overexpression[13,19,38,39]. This is due to a single-amino acid deletion, ΔN15, which destabilizes the overall protein structure[23]. Therefore, because *A3H* ΔN15 haplotypes III and IV are unlikely to have functional importance, we focused further computational

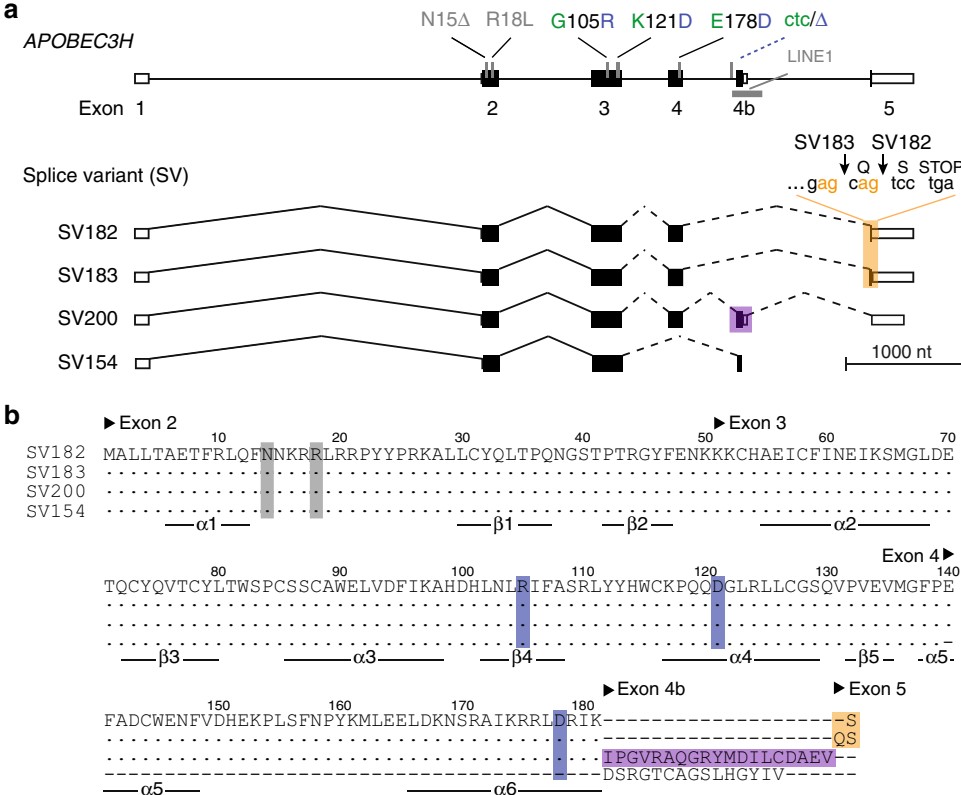

**Fig. 1** Human *A3H* genetic variation. **a** Schematic of the 5 exon human *A3H* gene with key coding and non-coding variations. Haplotype I and II variations are colored green and blue, respectively. The four different splice variants are shown below with orange shading to highlight the tandem splice acceptor that distinguishes SV182 and SV183 and purple shading to highlight the alternatively spliced exon 4b in SV200. **b** An amino acid alignment of each of the different splice variants encoded by the *A3H* haplotype II locus. To correspond with labels in **a**, variable amino acid positions are shaded gray or blue and alternative splicing products are shaded orange or purple. Exon positions are indicated above the amino acid alignment and secondary structures elements below

**Table 1 APOBEC3H haplotypes and frequencies from analyses of 1000 Genomes Project data (n = 2504)**

| APOBEC3H haplotype | Amino acid position and variant | | | | | Global allele frequency |
|---|---|---|---|---|---|---|
| | 15 | 18 | 105 | 121 | 178 | |
| I | N | R | G | K | E | 46.4 |
| II | N | R | R | D | D | 20.9 |
| III | Δ | R | R | D | D | 11.1 |
| IV | Δ | L | R | D | D | 19.1 |
| V | N | R | R | D | E | 0.36 |
| VI | Δ | L | G | K | D | 0.34 |
| VII | N | R | R | K | E | 0 |
| VIII | Δ | L | R | D | E | 0.56 |
| IX | N | R | G | D | E | 0.50 |
| X | Δ | R | R | D | E | 0.40 |
| XI | Δ | R | G | K | E | 0.12 |
| XII | Δ | L | G | K | E | 0.12 |
| XIII | N | R | G | K | D | 0.06 |

and biological studies on haplotypes I and II, which have global allele frequencies of 46.4 and 20.9%, respectively. Moreover, due to geographic biases in *A3H* haplotype distributions, average allele frequencies vary dramatically in different regions of the world with, for instance, haplotype I constituting 65% in Asia and haplotype II 57% in sub-Saharan Africa (Fig. 2a).

Human *A3H* has 4 known splice variants (SV) that translate into proteins with 182, 183, 200, and 154 amino acids[39] (Fig. 1a, b). Genetic polymorphisms often underlie alternative splicing

events[46–48]. We therefore asked whether *A3H* haplotypes I and II are spliced differentially. To answer this question, we first surveyed RNAseq read depths of *A3H* exons in EBV-immortalized lymphoblastoid cell lines (LCLs) from 20 homozygous haplotype I and 20 homozygous haplotype II donors selected randomly from the 1000 Genomes Project[45]. These RNAseq profiles revealed a major difference between mRNA species with respect to exon 4b (2 representative profiles in Fig. 2b and 40 in Supplementary Fig. 1). Exon 4b is mostly excluded from mRNA species of *A3H* haplotype I LCLs but is frequently included in mRNAs of *A3H* haplotype II LCLs. On average, exon 4b is expressed fivefold more frequently in *A3H* haplotype II versus haplotype I LCLs (15.8 ± 1.2 versus 3.0 ± 0.2; n = 20 LCLs representing each haplotype; p = 6 × 10⁻⁹, two-tailed Student's t-test).

These results predicted that homozygous *A3H* haplotype II individuals will express SV182, SV183, and SV200, whereas homozygous *A3H* haplotype I individuals will only express the former 2 splice variants. To test this prediction, Geuvadis Project[49] RNAseq data sets were analyzed for *A3H* haplotypes and mRNA levels. As predicted, LCLs from individuals homozygous for *A3H* haplotype I predominately express SV182 and SV183 at similar frequencies, each ~45% (Fig. 2c). In contrast, LCLs from individuals homozygous for *A3H* haplotype II express SV182, SV183, and SV200 at statistically indistinguishable frequencies, each ~32% (Fig. 2c). Moreover, *A3H* haplotype I and II heterozygous LCLs show intermediate levels of all three splice variants. These data strongly indicate that *A3H* differential splicing has a genetic basis, most likely due to one or more SNPs/indels within the locus itself.

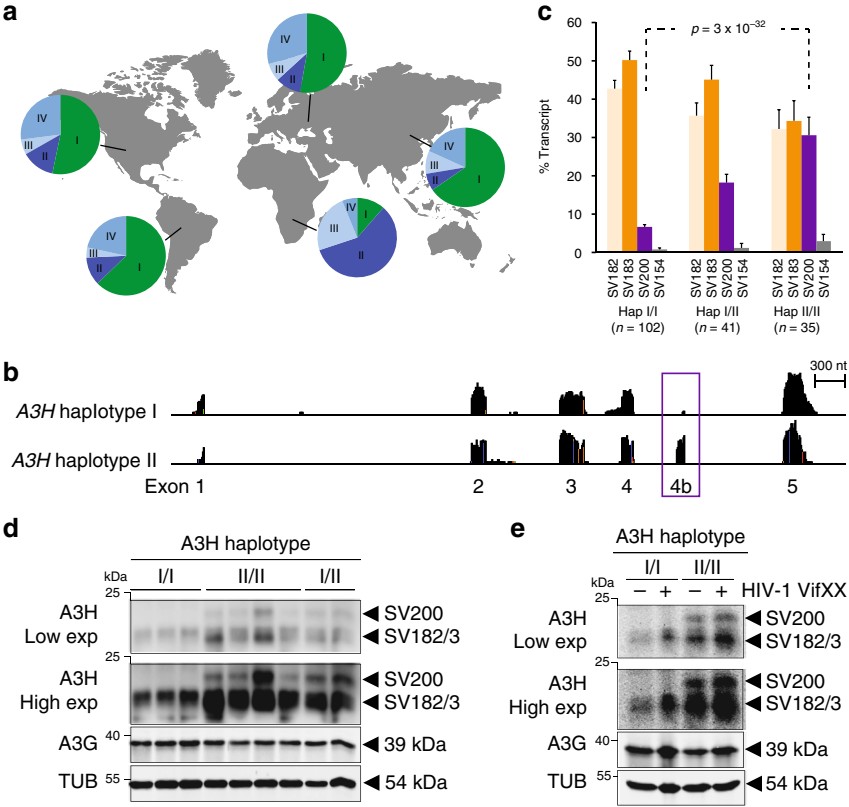

**Fig. 2** Differential splicing and global distributions of *A3H* haplotypes I and II. **a** Pie charts representing the frequency of each major *A3H* haplotype within human populations in the indicated geographic regions. **b** Representative RNAseq profiles of LCLs derived from individuals homozygous for *A3H* haplotypes I or II with the exon 4b region boxed. **c** Bar charts reporting the average percentage of each splice variant observed in LCLs from donors with the indicated *A3H* genotype (*n* values are shown and error bars represent 95% CI, *p* value is calculated using two-tailed Student's *t*-test). **d**, **e** Immunoblots of whole-cell extracts derived from LCL (**d**) or PBMC (**e**) with the indicated *A3H* haplotypes. PBMC were mock-treated or infected by Vif-deficient HIV-1 to induce *A3H* gene expression. Low and high exposures are shown for anti-A3H blots to show SV200 specificity to the haplotype II genotype. Anti-A3G and anti-TUBULIN (TUB) blots are shown as controls

The near equal expression of SV182 and SV183 (regardless of A3H haplotype) is most likely due to a splice acceptor slippage mechanism, in which the splicing apparatus has an equal chance of using AG dinucleotide splice acceptor sites arranged tandemly at the beginning of exon 5 (5′-gag-cag in Fig. 1a). Analogous acceptor slippage events have been documented for other genes, e.g., ref. [50]. The fourth splice variant, SV154, excludes exon 4 and includes exon 4b, and it is only observed at low levels regardless of *A3H* haplotype (<2–3%). Moreover, SV154 is predicted to encode a non-functional enzyme due to a loss of multiple conserved structural elements[23] (Fig. 1b). Therefore, studies hereon focus exclusively on SV182, SV183, and SV200.

***A3H* haplotype II expresses three distinct proteins**. To investigate whether these mRNA splicing differences manifest at the protein level, whole-cell lysates from LCLs homozygous for *A3H* haplotypes I and II were immunoblotted with a rabbit anti-A3H polyclonal antibody (Fig. 2d) and independently with a mouse anti-A3H mAb P1D8–1 that binds a shared N-terminal epitope[19] (Supplementary Fig. 2). Strong signals were evident for SV182/183 proteins in all *A3H* haplotype I and II LCLs, and a less intense but clearly detectable signal was observed for the SV200 enzyme only in LCLs with at least one *A3H* haplotype II allele (Fig. 2d). The same antibodies detected all three splice variants in Vif-deficient HIV-1 infected PBMCs from *A3H* haplotype II individuals, but only the shorter two splice variants in haplotype I specimens (Fig. 2e and Supplementary Fig. 2). Endogenous A3H haplotype I is expressed at lower levels than haplotype II in both

LCLs and PBMCs as reported[19] (compare SV182/3 band intensities for homozygous specimens in Fig. 2d, e). However, despite the technical challenge of lower expression levels, *A3H* haplotype I samples yielded clear signals for SV182/3 enzymes and no signal for SV200, even at highest exposure levels (Fig. 2d, e). Importantly, quantification of the amount of SV200 in immunoblots from homozygous haplotype II PBMCs and LCLs showed that this variant comprises 30–35 and 16–43%, respectively, of total A3H protein (*n* = 4 and *n* = 8 independent experiments, respectively; Fig. 2d, e and additional blots not shown). This result is concordant with SV200 constituting one-third of total A3H haplotype II splice products (Fig. 2c) and approximately one-third of total cellular A3H protein (Fig. 2d, e). Together, these data demonstrate that the SV200 enzyme is a major expressed splice variant with similar opportunities (based on abundance) to contribute to antiviral immunity as SV182 and SV183.

**SNPs associated with *A3H* exon 4b alternative splicing**. To investigate the mechanism responsible for *A3H* alternative splicing, we first tested the hypothesis that differential splicing may be due to one of the known coding SNPs G105/R105, K121/D121, and/or E178/D178 that distinguish *A3H* haplotypes I and II (Fig. 1). RNAseq data were used to quantify expression levels of all 4 *A3H* splice variants in donors with homozygous haplotype III or IV genotypes. Haplotypes III and IV are identical to haplotype II with respect to these three coding SNPs (Table 1). Accordingly, haplotypes III and IV LCLs would be predicted to

express SV200 at levels comparable to haplotype II LCLs. However, these analyses showed that haplotypes III and IV LCLs express much lower levels of SV200 mRNA indicating that none of these coding SNPs is responsible for alternative splicing and expression of SV200 (compare Fig. 2c and Supplementary Fig. 3).

We next asked whether a non-coding SNP or indel may be responsible and performed an unbiased association analysis of 5488 SNPs covering the full *A3* locus ($n = 461$; Supplementary Data 1). The top candidate was a 3 bp polymorphism upstream of exon 4b (ctc/$\Delta$, rs149229175; Fig. 3a). The correlation between ctc copy number and SV200 expression is highly significant ($P_{regression} = 8 \times 10^{-81}$, ANOVA; Fig. 3b). With few exceptions, likely due to rare recombination events, we found that the ctc deletion SNP is highly specific to *A3H* haplotype II. Cells homozygous for the ctc deletion (mostly also homozygous haplotype II) express the highest levels of SV200. In contrast, cells homozygous for ctc (mostly haplotypes I, III, and IV) have undetectable or low levels of SV200 expression. As expected, heterozygous cells express intermediate SV200 levels (Fig. 3b).

Phylogenetic analyses indicated that the non-coding ctc deletion ($\Delta$ctc) is a unique and defining feature of the *A3H* haplotype II block (Fig. 3c). This deletion is absent in other human *A3H* haplotypes and in the corresponding genomic DNA sequences of non-human primates, chimpanzees and rhesus macaques (Fig. 3c and Supplementary Fig. 4). Moreover, as expected, transcriptomic data from these species failed to show evidence for inclusion of exon 4b or expression of longer *A3H* transcripts (Supplementary Fig. 4). Interestingly, $\Delta$ctc is also present in archaic genomes (Neanderthals and Denisovans; Supplementary Fig. 4). Original studies by the Emerman and Malik groups deduced that the ancestral hominid *A3H* gene encoded a stable protein, and that two different unstable variants emerged independently in humans (R105G in haplotype I and

$\Delta$N15 in haplotypes III and IV)[13]. Our analyses indicate that either $\Delta$ctc emerged as a result of a similar independent event or was introduced to modern haplotype II humans through recombination with an archaic genome (see Discussion for possible mechanism and origins).

An additional feature of this region of the *A3H* locus conserved in primates is a 195 bp antisense LINE1 insertion, L1PA17[51], which spans the entirety of exon 4b (Fig. 3a). It is therefore tempting to propose that the intronic $\Delta$ctc may have been the trigger that enabled exonization of this portion of this LINE1 sequence and, simultaneously, fortification of human innate immune function by creating a new A3H enzyme variant with differential antiviral activity.

**Molecular mechanism of *A3H* exon 4b alternative splicing**. To directly test the hypothesis that the intronic ctc deletion promotes alternative splicing to exon 4b, we constructed an *A3H* minigene with *A3H* exons 1–4, intron 4 with and without ctc (both including exon 4b), and exon 5 including the natural 3′ untranslated region (Fig. 4a). Transient minigene expression in HeLa and MCF7 cell lines yielded all 3 splice variants including SV200 (Fig. 4b). SV200 was only expressed from the $\Delta$ctc construct demonstrating that this 3 bp deletion is essential for exon 4b inclusion and SV200 expression. The deleted ctc is part of an extended polypyrimidine tract, 5′-ttctcctctc-3′, which is a predicted binding site for several splicing factors including PTBP1, U2AF2, hnRNP C, and SRSF5[52–58]. A reduction from 10 to 7 consecutive pyrimidines is likely to change the repertoire of splicing factors bound to this site, influence interactions with other splice factors, and enable inclusion of exon 4b or, alternatively, weaken an exon 4b exclusion mechanism. It is additionally notable that the ratio of SV200 to SV182/3 protein produced from the minigene is lower than that of the same

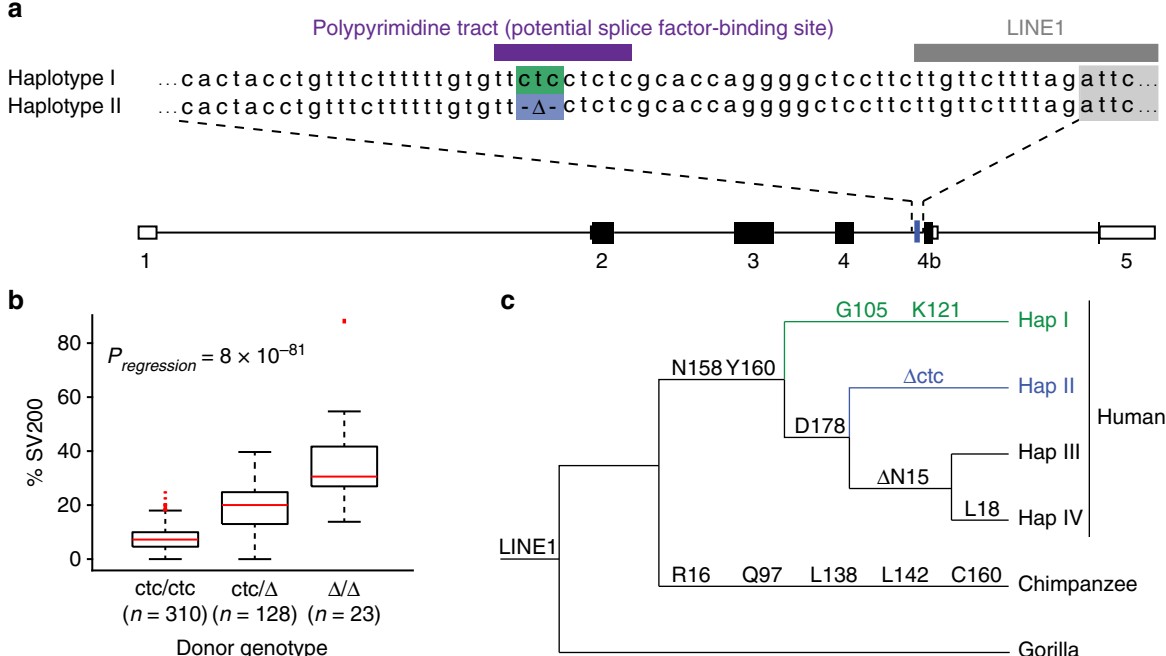

**Fig. 3** An intronic $\Delta$ctc associates with A3H haplotype II and SV200 expression. **a** *A3H* intron 4 alignment with ctc in haplotype I (green) and $\Delta$ctc in haplotype II (blue) (rs149229175). Additional relevant *cis* elements include a polypyrimidine tract (purple) and a LINE1 (gray). **b** Box and Whisker plot showing the percentage of SV200 transcripts in cells with the indicated *A3H* intron 4 genotypes. In this plot, boxes, middle lines and broken lines display interquartile ranges (IQRs), medians, and 1.5 IQR, respectively. Outliers (>1.5 IQR) are shown and the p value is obtained using one-way ANOVA. **c** A schematic showing the phylogenetic relationships between the major human *A3H* haplotypes (frequencies >1%) and selected non-human primate homologs. The LINE1 insertion, coding changes, and the haplotype II-specific ctc deletion are indicated in relevant divergence intervals (not to a specific time scale)

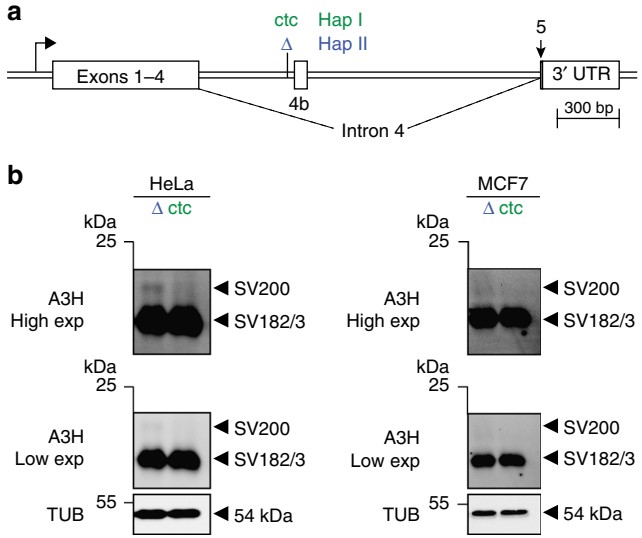

**Fig. 4** Influence of non-coding intron 4 ctc versus Δctc on *A3H* minigene expression. **a** Schematic of the *A3H* minigene construct with key elements labeled. The minigene haplotype I intron has ctc (green) and the haplotype II intron has Δctc (blue). These constructs are otherwise identical. **b** Immunoblots of HeLa and MCF7 whole-cell extracts expressing the indicated *A3H* minigene constructs (representative of two independent experiments). Low and high exposures are provided to show that SV200 expression is specific to the haplotype II minigene. An anti-TUBULIN (TUB) blot is shown as a loading control

variants produced from the chromosomal locus. This difference may be due to the context of the gene (episomal versus chromosomal) and/or to other contributing cis/trans-acting factors such as chromatin states, minor SNPs, and splice factors.

**The discovery of SV200 cleavage by viral protease.** To directly compare the HIV-1 restriction capabilities of untagged A3H haplotype II SV182, SV183, and SV200, dose–response experiments were done with varying amounts of each splice variant against Vif-deficient HIV-1 (strain LAI)[23]. All three splice variants expressed well in 293T cells as evidenced by similar immunoblot band intensities (Fig. 5a). All three A3H splice variants also packaged into virus particles and, like A3G, elicited a dose-dependent restriction of virus infectivity (Fig. 5a). However, A3H SV200 was invariably found in two distinct forms in viral particle immunoblots, despite being monomorphic in whole-cell lysates (Fig. 5a). The slower migrating band appeared at the expected molecular weight for the full 200 residue protein (23 kDa), and the virus particle-specific faster migrating band appeared between those of SV182/3 (21 kDa) and SV200.

These unexpected results suggested that a protease specifically cleaves SV200 inside HIV-1 particles. To test this possibility, Vif-deficient HIV-1 restriction experiments were done with SV183 and SV200 in the absence or presence of varying amounts of darunavir (DRV) or atazanavir (ATV), which are potent and highly selective HIV-1 protease inhibitors (Fig. 5b). Following a PBS wash 6 h post-transfection, protease inhibitor or vehicle control was added to the virus-producing cells and after an additional 42 h of incubation virus-containing supernatants were collected for analysis. Neither protease inhibitor affected cellular or viral levels of A3H-II SV183, but both drugs clearly prevented, in a dose-dependent manner, the accumulation of the smaller-sized form of A3H-II SV200 in viral particles. The protease inhibitor treatments were robust as they also prevented Gag

processing and suppressed virus infectivity. Taken together, these data demonstrated that HIV-1 protease specifically cleaves A3H SV200 into a smaller form within viral particles.

Cleavage of A3H haplotype II SV200 by HIV-1 protease is most likely occurring near the C-terminus between residues 183 and 200 because this polypeptide region is unique to this splice variant (i.e., the N-terminal 181 residues are shared between all three splice variants and SV182 and SV183 are unaffected in viral particles; alignment in Fig. 1 and immunoblots in Fig. 5). The HIVcleave webserver[59] predicted a high-confidence cut site between amino acid positions 186 and 187 within the unique C-terminal extension of A3H SV200. To test this prediction, single-alanine mutants of SV200 spanning residues 184 to 190 were used in virus particle immunoblot experiments (excluding 187 which is naturally alanine). Most of these single-amino acid changes showed altered rates of proteolytic cleavage with the strongest phenotypes being hypo-cleavage (V185A) and hyper-cleavage (R186A; Fig. 5c). Additional single-amino acid substitution mutants and a panel of SV200 truncation mutants (stop codons at positions 185, 186, 187, and 188) provided further support for HIV-1 protease acting at this site (Supplementary Fig. 5). In addition, A3H haplotype II SV200 cleavage by HIV-1 protease is likely to be a conserved function because representative lab strains (subtype B) and clinical HIV-1 isolates from subtypes B and C (transmitted founders and chronic circulating isolates), but not SIVmac239, showed the same cleavage event within particles (Fig. 5d).

**A3H-II SV200 hyperactivity is attenuated by HIV-1 protease.** Prior studies comparing the HIV-1 restriction activities of the different human A3H splice variants were constrained by a need to use epitope tags and additionally limited because A3H amounts and particularly SV200 levels were not quantified per unit incorporated into viral particles[39,42,44,60]. Moreover, experiments above in Fig. 5 indicated similarly strong Vif-deficient HIV-1 restriction activities for SV182/3 and SV200 but the activity of the latter splice variant was likely underestimated due to lower enzyme amounts in viral particles (and potentially also to proteolytic cleavage).

We therefore sought to quantify the Vif-deficient HIV-1 restriction activities of untagged A3H haplotype II SV200, a hypo-cleavable mutant SV200 V185Q, a hyper-cleavable mutant SV200 R186A, and SV183 in a series of single-cycle experiments (mutants from Fig. 5c and Supplementary Fig. 5). In these titration experiments, restriction activities were quantified as a function of the amount of encapsidated A3H normalized to p24 (Fig. 6 and Methods). Interestingly, A3H SV200 and the hypo-cleavable SV200 derivative V185Q showed the strongest restriction activities per unit of encapsidated protein, whereas SV183 showed the weakest and the hyper-cleavable variant R186A showed intermediate levels (quantification in Fig. 6a of immunoblots from three biologically independent experiments shown in Fig. 6b). In quantitative terms, A3H SV200 is fourfold more restrictive than SV183 and at least twofold more restrictive than the hyper-cleavable variant R186A (Fig. 6). Thus, altogether, these data indicate that HIV-1 protease functions to attenuate the potent hyper-restriction activity of A3H SV200.

**Discussion**

*A3H* is the most variable human *A3* gene with five biallelic amino acid positions that make-up at least 12 different haplotype blocks (including 6 new ones reported here). Additional *A3H* diversity occurs through four different splice variants. Here we show that the splice variant encoding the longest protein, SV200, is expressed as a result of a non-coding 3 nucleotide deletion (Δctc)

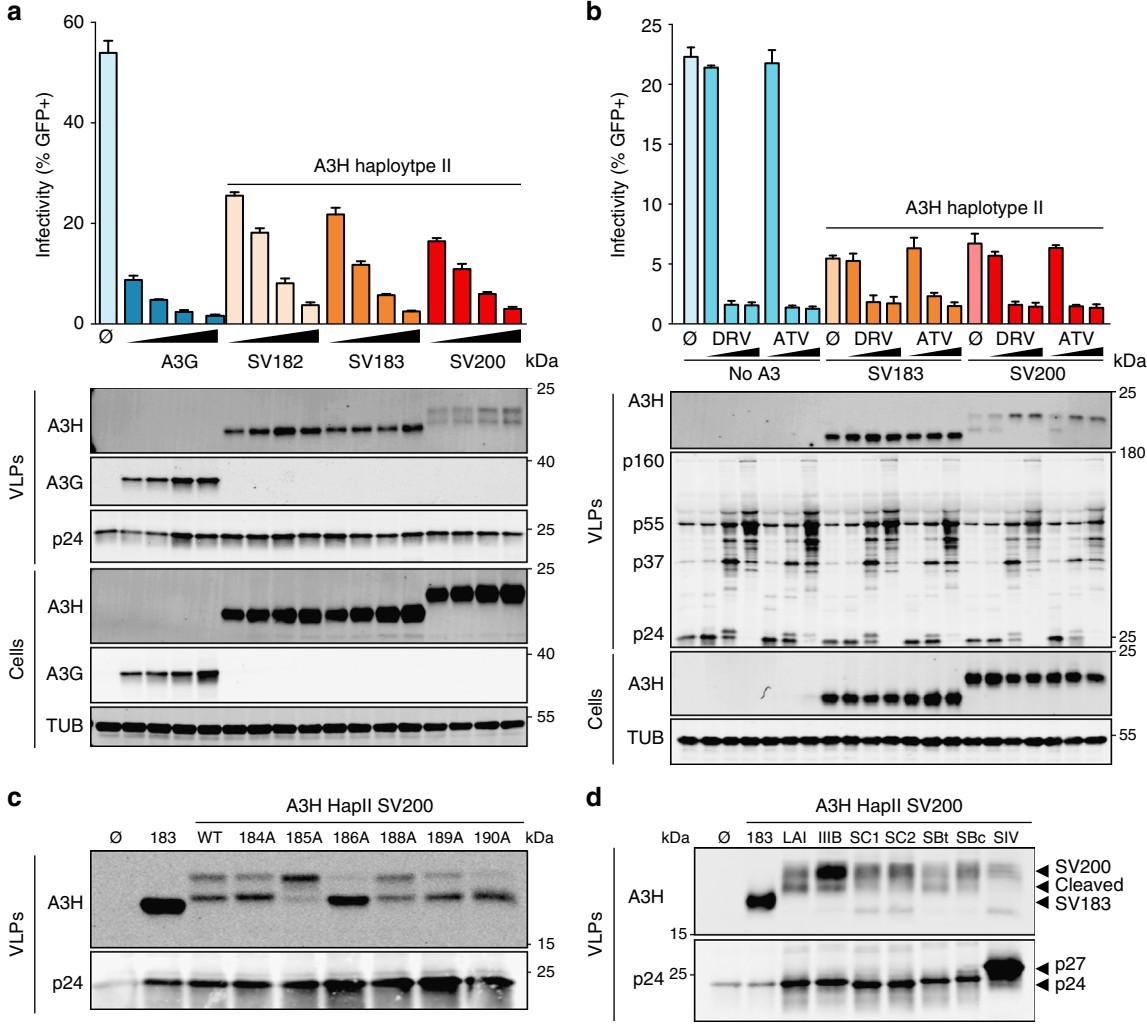

**Fig. 5** HIV-1 restriction by A3H haplotype II splice variants and SV200 processing by HIV-1 protease. **a** Infectivity of Vif-deficient HIV-1<sub>LAI</sub> produced in 293T cells in the presence of untagged A3H splice variants (50 ng, 100 ng, 200 ng, or 400 ng) or V5-tagged A3G (25 ng, 50 ng, 100 ng, or 200 ng) and used to infect CEM-GFP reporter T cells ($n = 3$ technical replicates; average ± SD). Immunoblots are shown below for producer cells and virus-like particles (VLPs). **b** Infectivity of Vif-deficient HIV-1<sub>LAI</sub> produced in 293T cells in the presence of a control vector (no A3) or the indicated untagged A3H splice variants (200 ng). HIV-1 protease inhibitors (0.3 nM, 3 nM, or 300 nM) darunavir (DRV) or atazanavir (AZV) were added 6 h post-transfection or 42 h before virus particle collection, virus infectivity assays using CEM-GFP reporter cells, and VLP and cell lysate immunoblotting as in (**a**). **c** Immunoblots of A3H haplotype II SV200 and mutant derivatives in VLPs, following production in 293T cells. Gag (p24) is shown as a loading control. **d** Immunoblots of A3H haplotype II SV183 and SV200 in VLPs from the indicated proviral plasmids, following production in 293T cells. Gag (p24 for HIV-1 isolates and p27 for SIV<sub>mac239</sub>) is shown as a loading control

unique to the haplotype II. The other major splice variants, SV182/3, occur at similar frequencies in all major haplotypes and are therefore not haplotype-specific. Quantification of Vif-deficient HIV-1 restriction activity showed that SV200 is at least fourfold more potent than that of the shorter splice variants. Thus, the A3H haplotype II-specific Δctc is the first non-coding polymorphism shown to change the amino acid composition and affect the function of a human APOBEC enzyme. Furthermore, functional characterization of this splice variant led to the surprising finding that HIV-1 protease specifically cleaves SV200 into a shorter 186 amino acid isoform. Additional Vif-deficient HIV-1 restriction experiments with hypo-cleavable and hyper-cleavable single-amino acid mutant derivatives of A3H haplotype II SV200 indicated that this cleavage event attenuates antiviral activity. Thus, these studies combined to show that HIV-1 protease provides an additional layer of protection against restriction by A3H haplotype II, in addition to the established Vif-mediated proteasomal degradation mechanism.

*A3H* is the only *A3* gene with a repetitive element that contributes to the coding sequence (LINE L1PA17, Fig. 1a and Fig. 3a). Phylogenetic comparisons of human and non-human primate *A3H* genes indicate that this insertion occurred early in the primate lineage, most likely prior to the evolutionary divergence of known primates (Fig. 3c). This conclusion is supported by a lack of a homologous LINE1 insertion in the *A3Z3* gene (*A3H* homolog) of non-primate mammalian lineages (DNA sequence alignments not shown). The specificity of the intronic Δctc to *A3H* haplotype II (apart from rare recombinants) and the highly significant correlation between the occurrence of this deletion variant and alternative splicing to include exon 4b and ultimately produce the A3H haplotype II SV200 enzyme (Figs. 2c, 3b) combine to support a model in which the deletion itself may have been the final genetic step in creating exon 4b from LINE1 sequence (i.e., birth of a new exon). Further work will be needed to determine if the Δctc promotes an exon inclusion mechanism or suppresses an exon exclusion mechanism because

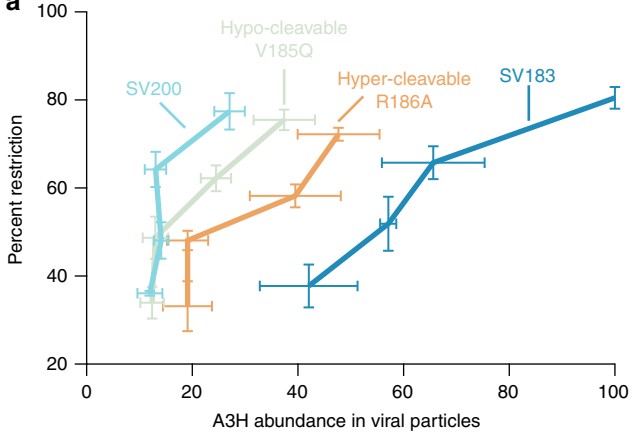

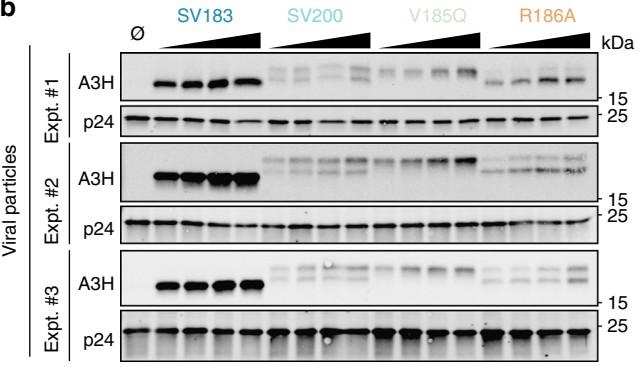

**Fig. 6** Quantification of HIV-1 restriction for A3H haplotype II SV200 and hypo- and hyper-cleavable derivatives. **a** Dose–response summary of infectivity data for Vif-deficient HIV-1$_{LAI}$ produced in 293T cells in the presence of the indicated untagged A3H splice variants (12.5 ng, 25 ng, 50 ng, or 100 ng) and infectivity quantification by flow cytometry of infected CEM-GFP reporter T cells. Each data point represents the mean of three biologically independent experiments. The percent restriction values (y-axis) were obtained by normalization to no A3 control reactions (mean ± SEM). Relative A3H abundance in viral particles (x-axis) was determined by normalization to p24 levels (mean ± SEM; see main text and methods for additional details). **b** Immunoblots of A3H and p24 in virus particle lysates for the three biologically independent experiments quantified in (**a**)

there are clear precedents for both in the literature[52–56]. However, to our knowledge, this is the first example of LINE1 exonization of a primate *A3* gene. This is ironic given that A3H has been shown to elicit strong LINE1 restriction activity[13,61,62].

DNA polymerases are known sources of indels, particularly in repetitive sequence tracts[63]. The original Δctc may therefore have been caused by a DNA polymerase slippage event that removed one of the three possible ctc trinucleotides from the larger poly-pyrimidine tract 5′-tt(ctc)(ct(c)tc)-3′ (brackets demark possibilities). The timeframe in which this event occurred is difficult to pinpoint but most likely prior to the global radiation of the four most common *A3H* haplotypes (I, II, III, and IV; Fig. 2a). The original Δctc may have independently occurred one or more times in an ancestral *A3H* haplotype II population, as it does not exist (apart from rare recombinants) in other modern human *A3H* haplotype populations. An alternative hypothesis is that Δctc emerged in modern hapII humans as a result of recombination with an archaic genome. An analogous Δctc variant exists in Neanderthal and Denisovan genomic DNA sequences (Supplementary Fig. 4). Moreover, within the 7 kb *A3H* gene there are an additional 13 SNPs specific to archaic and modern humans with the haplotype II genotype. In contrast only one SNP is

shared between archaic and modern humans with the haplotype I genotype. Altogether, these data suggest that Δctc may have an archaic origin. However, further studies are needed to pinpoint the exact source of this important variant.

Regardless of the exact window in time that the Δctc variant entered the human *A3H* haplotype II population, it was most likely long before HIV-1 became a pandemic virus (i.e., multiple thousands of years ago versus hundreds of years ago for HIV-1 zoonosis[64]). This suggests that the antiviral activity of A3H haplotype II SV200 is not limited to HIV-1 and may be considerably broader, consistent with the current working model for A3-mediated innate immunity in which multiple enzymes combine to form a broad, overlapping defense (reviewed by refs. [2,3]). We speculate that the emergence of Δctc in ancestral humans may have originally conferred resistance to an ancestral human pathogen. Such resistance may have conferred a selective advantage to ancestral A3H haplotype II humans and promoted their enrichment in the overall population and particularly in sub-Saharan Africa where *A3H* haplotype II is currently dominant (57% allele frequency equating to 82% of the population with at least 1 copy, versus other parts of the world such as a 7–10% allele frequency in Asians and Europeans, respectively equating to 13–19% of the population with at least one copy; Fig. 2a). Haplotype-specific differential splicing has the potential to have a significant contemporary global impact because haplotype II individuals will have a larger, and potentially stronger, antiviral A3H repertoire, overall A3 protein repertoire, and therefore potential to restrict a broader array of pathogens. It is not hard to imagine a scenario, for instance, in which populations with *A3H* haplotypes I, III, and IV may be more susceptible to a virus (or DNA-based parasite) than those with *A3H* haplotype II. *A3H* heterozygotes may have an even greater advantage given significant amino acid and functional differences between haplotypes.

The increased potency of A3H haplotype II SV200 against Vif-deficient HIV-1 and counteraction by HIV-1 proteolytic cleavage strongly suggests that this variant enzyme poses a significant threat to the virus. It is tempting to speculate that the viral protease-dependent A3 counteraction mechanism may have been part of the overall process of HIV-1 adaptation to the human population (especially given the dominance of this allele in sub-Saharan Africa corresponding with the likely geographical origin HIV-1 in humans[64]). This idea is supported conservation of this activity amongst multiple different HIV-1 strains/sub-types but not for the rhesus macaque virus, SIV$_{mac239}$ (Fig. 5d). Moreover, it may be a human-specific fortification of the conserved Vif-dependent A3 degradation mechanism because the Δctc variant, intron 4b inclusion, and SV200 production are all unique features of the human *A3H* gene that do not occur in non-human primate *A3H* genes. Although this is the first example of a human retroviral protease cleaving a human A3 enzyme, there are at least two other reports of a retroviral protease cleaving a host A3 enzyme (i.e., FIV protease cleaving feline A3[65] and MLV protease cleaving murine A3[66]). Thus, we propose that viral proteases may provide an additional layer of protection against restriction by A3 enzymes and, depending on the virus and host species, may constitute either a primary or a secondary counteraction mechanism.

## Methods

**Bioinformatic analyses.** We first determined the *A3H* genotype of 2054 individuals from the genetic polymorphism data set of the 1000 Genomes Project, Phase 3. 461 of these individuals were represented by RNAseq data reported by the Geuvadis Project[49]. Using these two data sets we identified and randomly picked 20 homozygous *A3H* haplotype I and 20 homozygous haplotype II and inspected their RNAseq BAM files in the Integrative Genomics Viewer (IGV) tool (Supplementary Fig. 1). To compare the expression of exon 4b in these 40 samples, we quantified

the read depth of this exon defined as the sum of read depths at all bases of exon 4b normalized by the read depth of exon 3 for each sample. The average percentage read depth of exon 4b was 15.8 with a 95%CI of 1.2 in haplotype II cells and 3.0 with a 95%CI of 0.2 in haplotype I cells.

Using the SAMtools program[67] we sorted and indexed the RNAseq BAM files and then used the Cufflinks program[68] to assemble *A3H* transcripts and estimate their abundance in each individual. The information of known *A3H* transcripts was provided to the Cufflinks program as a GFF3 file containing ENSMBL annotated *A3H* transcripts ENST00000421988 (SV-154), ENST00000348946 (SV-182), ENST00000442487 (SV-183), and ENST00000401756 (SV-200). The output of this analysis for each *A3H* transcript of each individual is a FPKM (Fragments Per Kilobase of transcript per Million mapped reads), which is a measure of the abundance of each transcript. We presented these expression data as percentages such that the sum of FPKM values of all *A3H* transcripts within an individual sample is equal to 100.

To identify the genetic source of differential *A3H* haplotype II SV200 expression, we performed a global correlation analysis between the expression of SV200 in 461 donors (Geuvadis Project LCLs) and the number of copies of each allele in each of the 5488 polymorphic positions in the *A3* locus (human chromosome 22 region spanning *A3A-A3H* plus 1000 nucleotides upstream of *A3A* and downstream of *A3H*).

A phylogenetic analysis was done using a multiple alignment of full-length *A3H* gene sequences from human haplotypes I (hg reference), II, III, IV, chimpanzee, and gorilla. The *A3H* genes of each of these species as well as that of rhesus macaque were extracted directly from the UCSC Genome Browser or deduced from known variants using the 1000 Genomes Project Database. The Neanderthal and Denisovan *A3H* sequences were inferred from their corresponding UCSC Genome Browser tracks.

**Endogenous A3H immunoblots.** LCLs were obtained from the NIGMS Human Genetic Cell Repository at the Coriell Institute for Medical Research. One archived PBMC specimen used in Supplementary Fig. 2 came from a HIV-1 uninfected Ugandan participant in the Partners in Prevention HSV/HIV Transmission Study[69]. Regulatory review for this is covered by protocol review and approval through institutional review boards of the Ugandan HIV/AIDS Research Committee (NARC), the Ugandan National Council for Science and Technology (UNCST), and the University of Washington (UW), with the University of Minnesota added through modification of the UW approval. Additional PBMC specimens were isolated from mononuclear cells enriched Trima Cones (Memorial Blood Centers, St. Paul, Minnesota) using Ficoll-Paque Premium (GE Healthcare). These additional specimens were classified as exempt from review by the University of Minnesota Institutional Review Board (IRB) because donors are deidentified by Memorial Blood Centers. Cells were stimulated with 0.5 μg/mL anti-CD3 antibody (clone UCHT1, R&D System), 0.1 μg/mL anti-CD28 antibody (clone CD28.2, eBioscience) and 20 U/mL interleukin-2 (IL-2) (Miltenyi Biotec) in R20 medium (RPMI with 20% FBS and 1% P/S) for 3 days. Fluorescein (FITC) conjugated antibodies against CD8 and CCR7, as well as Phycoerythrin (PE) conjugated antibodies against CD4 and CD45RO were used to verify composition and stimulation of cells. PBMCs were maintained in R20 medium with 20 U/mL IL-2. Vesicular stomatitis virus G protein (VSV-G) pseudotyped viruses were generated by transfecting 8 μg of proviral HIV-1 IIIB VifX26 × 27 (Vif-deficient) expression construct and 2 μg of VSV-G expression construct into 293T cells. Viral titers were quantified using the CEM-GFP reporter cell line[27]. PBMCs were infected at a 0.5 MOI and harvested 10 days post-infection. LCLs, non-infected, and infected PBMCs, respectively, were lysed and subjected to immunoblot analysis. For immunoblotting, cells were pelleted, washed with 1× PBS, and then lysed with 2.5× Laemmli sample buffer. Lysates were then subjected to SDS-PAGE followed by protein transfer to PVDF using a Bio-Rad Criterion system. Membranes were probed with a mouse anti-A3H mAb P1D8[19], a rabbit anti-A3H pAb (NBP1–91682, Novus), a rabbit anti-A3G (NIH AIDS Reagent Program 10201 courtesy of Dr. Jaisri Lingappa), and a mouse anti-tubulin mAb (Covance). Secondary antibodies were goat anti-rabbit IRdye 800CW (Li-COR 926–32211) and goat anti-mouse Alexa Fluor 680 (Molecular Probes A21057). Additional secondary antibodies were donkey anti-Ms IgG-HRP (Jackson Laboratory) and goat anti-Rb IgG-HRP (Bio-Rad). All primary antibodies were used at a 1:1000 dilution, and all secondary antibodies were used at a 1:10,000 dilution. Membranes were imaged using commercial chemiluminescence reagent (Thermo Scientific™ SuperSignal™ West Femto Maximum Sensitivity Substrate Cat#34095) and a LI-COR Odyssey instrument. Images were prepared for presentation using ImageJ. Uncropped immunoblot images are included as Supplementary Fig. 6.

***A3H* minigene alternative splicing experiments.** To make the *A3H* minigene constructs, genomic DNA was extracted from a homozygous *A3H* haplotype II LCL sample (ID: HG00442; obtained from NIGMS Human Genetic Cell Repository at the Coriell Institute for Medical Research) and *A3H* intron 4 was amplified by high-fidelity PCR, cloned into pJET1.2, and verified by DNA sequencing. Next, a gBlock was digested with *Xma*I/*Nco*I and ligated into the pJET1.2 *A3H* haplotype II intron 4 construct to introduce a unique *Bam*HI site into *A3H* exon 3, which was then confirmed by DNA sequencing. A 'ctc' insertion to restore the intronic

polypyrimidine tract was then performed using a standard site-directed mutagenesis protocol on the pJET1.2 *A3H* haplotype II intron 4 construct. Both constructs (±ctc) were then digested with *Bam*HI/*Xba*I, and cloned into the base *A3H* haplotype II expression vector. Subsequently, the 3′-UTR from the HG00442 LCL genomic DNA was amplified, ligated into pJET1.2, and confirmed by DNA sequencing. The 3′-UTR was then added to both of the expression constructs (±ctc) using compatible *Ale*I/*Xba*I cut-sites.

One microgram of each *A3H* minigene containing plasmid was transfected into either HeLa or MCF7 cell lines (ATCC) using a 3:1 (reagent:DNA) ratio of TransIT LT1 (Mirus). Forty-eight hours post-transfection, cells were trypsinized, pelleted, washed with 1× PBS, and then lysed with 2.5× Laemmli sample buffer. Lysates were subjected to SDS-PAGE, protein transfer, antibody incubations, and subsequent imaging as described above. Primary antibodies were a rabbit anti-A3H pAb (NBP1-91682, Novus) and a mouse anti-tubulin mAb (Covance). Secondary antibodies were goat anti-rabbit IRdye 800CW (Li-COR 926–32211), goat anti-mouse Alexa Fluor 680 (Molecular Probes A21057), and goat anti-Rb IgG-HRP (Bio-Rad). All primary antibodies were used at a 1:1000 dilution, and all secondary antibodies were used at a 1:10,000 dilution. The blots were imaged on a bi-capable Li-Cor Odyssey chemiluminescent/fluorescent scanner.

**HIV-1 packaging and restriction experiments.** The HIV-1$_{LAI}$ VifX26 × 27 proviral construct has been described[23]. Additional HIV-1 molecular clones include a Vif-deficient subtype B lab strain (IIIB), Vif-proficient subtype B clinical isolates (CH58 T/F, ARP #11856, GenBank JN944907; RHGA, ARP #12421, GenBank KC312510-KC312539, and KC312467-KC312509), Vif-deficient subtype C clinical isolates (Z3618M T/F, ARP #13262, GenBank KR820366; Z3618F, ARP #13263, GenBank KR820342), and SIVMAC239 SpX (ARP#12249, Genbank M33262). The HIV-1 subtype C isolates were rendered Vif-deficient by introducing tandem stop codons at Vif positions 26 and 27 by subcloning the vif fragment into pJET1.2 vector, performing site-directed mutagenesis (Stratagene), and shuttling the fragment back into the original vector using BlpI and NcoI restriction sites. HIV-1 packaging and restriction experiments were done by transfecting (TransIt, Mirus) 50% confluent 293T cells (Harris lab collection) with 1 μg proviral plasmid DNA and each A3 expression construct or vector control as indicated in the relevant figure legend. After 48 h incubation, viral supernatants were cleared by filtration (0.45 μm) and used to infect CEM-GFP cells to monitor infectivity via flow cytometry.

Cell and viral particle lysates were prepared for immunoblotting as follows. Cells were pelleted, washed with 1× PBS, and lysed with 2.5× Laemmli sample buffer. Virus-containing supernatants were filtered and pelleted via centrifugation through a 20% sucrose cushion, then lysed in 2.5× Laemmli sample buffer. Lysates were then subjected to SDS-PAGE followed by protein transfer to PVDF using a Bio-Rad Criterion system. Membranes were probed with a mouse anti-A3H mAb P1D8[19], a rabbit anti-A3H pAb (NBP1–91682, Novus), a mouse anti-V5 mAb (Invitrogen), a mouse anti-HIV-1 Vif (NIH AIDS Reagent Program 6459 courtesy of M. Malim), an anti-HIV-1 p24/CA mAb (NIH AIDS Reagent Program 3537), or a mouse anti-tubulin mAb (Covance). Secondary antibodies were goat anti-rabbit IRdye 800CW (Li-COR 926–32211) or goat anti-mouse Alexa Fluor 680 (Molecular Probes A21057). An additional secondary goat anti-Rb IgG-HRP pAb (Bio-Rad) was also used. All primary antibodies were used at a 1:1000 dilution, and all secondary antibodies were used at a 1:10,000 dilution. Membranes were imaged using commercial chemiluminescence reagent (Thermo Scientific™ SuperSignal™ West Femto Maximum Sensitivity Substrate Cat#34095) and a LI-COR Odyssey instrument. Images were prepared for presentation using ImageJ. Uncropped immunoblot images are included as Supplementary Fig. 6.

Quantification of the levels of A3H in virus particles was performed using ImageJ (v1.47). First, viral particle A3H and Gag (p24) levels were quantified. Next, abundance of A3H in particles was determined by dividing the amount of A3H by its corresponding amount of Gag. Then, the highest abundance of A3H was set to 100 (A3H SV183 in this case; hence no horizontal error bars for this sample only) and all other A3H abundances were normalized against this value.

## Data availability

All A3H genomics and transcriptomic data sets used in this manuscript are available at 1000 Genomes (http://www.internationalgenome.org/data) and Jeuvadis (http://www.geuvadis.org) databases. The RNAseq data sets of non-human primates are available at the Nonhuman Primate Reference Transcriptome Resource (NHPRTR, www.nhprtr.org). A3H sequences of Denisovans and Neanderthals and non-human primates are available at the USCS Genome Browser (https://genome.ucsc.edu/). A full list of all primers used in this study is provided as Supplementary Table 1. The authors declare that all other data supporting the findings of this study are available within the article and its Supplementary Information files, or are available from the authors upon request.

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

## Acknowledgements

We thank Brett Anderson and Matthew Jarvis for comments, Kate Lauer for technical assistance, and Bruce Torbett for support. The following reagents were obtained through the NIH AIDS Reagent Program, Division of AIDS, NIAID, NIH: anti-human APO-BEC3G C-terminal polyclonal antibody (pAb 10201) from Dr. Jaisri Lingappa, pCH058. c/2960 (cat# 11856) from Dr. John Kappes and Dr. Christina Ochsenbauer, pRHGA (cat#12421) from Dr. Beatrice Hahn, HIV-1 Z3618M T/F (cat# 13262) and Z3618F Z3618F (SGA 11) infectious molecular clones (cat# 13263) from Dr. Eric Hunter, and SIV$_{mac}$239 SpX from Dr. Ronald C. Desrosiers. This work was supported by NIAID R37 AI064046 and NCI R21 CA206309 (to R.S.H.) and a Collaborative Development Project sub-award from NIGMS 2U54GM103368 and NIAID R21 AI138793 (to D.E.). Funding for the Partners in Prevention HSV/HIV Transmission Study was provided by the Bill and Melinda Gates Foundation, Grant # 26469. C.M.R. received salary support from NIAID T32-AI83196 and a University of Minnesota Doctoral Dissertation Fellowship, A. Z.C. from University of Minnesota Medical Scientist Training Program (NCI F30 CA200432) and NIGMS T32 GM008244, J.W. from a University of Minnesota Graduate School Interdisciplinary Doctoral Fellowship, J.L.M. from NSF Graduate Research Fellowship, D.J.S. from University of Minnesota Craniofacial Research Training (Min-nCResT) program (NIH T90DE022732), and G.J.S. from NSF Graduate Research Fellowship. R.S.H. is the Margaret Harvey Schering Land Grant Chair for Cancer Research, a Distinguished University McKnight Professor, and an Investigator of the Howard Hughes Medical Institute.

## Author contributions

Conception and design: D.E., C.M.R., and R.S.H. Acquisition of data: D.E., C.M.R., M.A. C., A.Z.C., J.T.B., J.W., T.I., J.L.M., and N.M.S. Analysis and interpretation of data: D.E., C.M.R., M.A.C., D.J.S., J.Y., and R.S.H. Technical and logistical support including training: D.J.S., G.J.S., J.R.L., J.Y., and W.L.B. Drafting manuscript: D.E., C.M.R., and R.S. H. Proofreading and approval of the final manuscript: All authors.

## Additional information

**Competing interests:** The authors declare no competing interests.

