## [Peer Review File · Nature Communications]

Reviewers' comments:

Reviewer #1 (Remarks to the Author):

Ebrahimi et al show that the most antiviral of the APOBEC3H haplotypes, haplotype II, also encodes a splice variant that creates a new exon. This splice variant makes a larger protein with increased antiviral activity, and is cleaved by HIV protease. While the data looks sound, there is some hyperbole related to its interpretation.

Major concerns

1. The major problem of the interpretation that the APOBEC3H SV200 splice variant has important antiviral activity, is that there is very little of it made relative to the other splice variants. Figure 2D is a good example of this where there looks like there is much less of the large band relative to the other bands that react with the antibody. Given how APOBEC3 proteins have dose-dependent effects, I am not certain that this splice variant really has that much of a role.
2. Figure 2E is confusing to me because the APOBEC3H bands increase in intensity after HIV infection. Shouldn't the Vif protein degrade the intracellular pools of APOBEC3H? In any case, showing if Vif has a differential effect on SV200 (i.e. that perhaps SV200 is more resistant to Vif) would add to the interest in this splice variant.
3. The authors hypothesize that the 3bp deletion that gives rise to the splice variant is human-specific. However, I did not see where they compared the intron sequences to chimpanzees to determine if this is true or not. This impacts much of their discussion about the adaptation of HIV to humans.
4. Similar to comment 3, the authors hypothesize that HIV-1 protease adapted to counteract the SV200 variant. However, they do not test this hypothesis by determining if other related lentiviruses, e.g. SIVmac or better SIVcpz, fail to cleave SV200 when this variant is co-transfected with the appropriate proviruses. A null hypothesis is that adaptation of protease did not occur, but just by chance the SV200 variant has a structure that just happens to be similar to one recognized by protease.
5. Try as I might, I just do not understand Figure 6. For one thing, there seems to be no dose-response in the expression of most of the APOBEC3H proteins. This would seem to indicate that they are already at saturation at the lowest doses, in which case, the calculations of abundance are not meaningful. Also, the Y axis of percent restriction is such a narrow range (40 to 80, so only a 2-fold range). I think this experiment needs to be done within a large linear range, and again, as in point #1, I am concerned that the levels of the splice variant relative to the major variant are too low to be biologically important.

Minor concerns

The presentation contains some over-enthusiastic writing, including

Page 3, top: Yes, bats with 13 genes is a lot, but is not "incredible." Humans have 14 interferon alpha genes, for example.

Page 3, top: No, the number of SNPs and insertions/deletions in APOBEC3 genes is not especially profound compared to many other genes. Also, it is important not to confuse this with positive selection as the text seems to do (ie. one is within a species, the other is between species).

Page 3, middle: "...there is a continuous battle.." is a little over-dramatic for a scientific paper.

Page 11, bottom: No, this is not the first description of a coding region formed from an L1 element. See <https://www.ncbi.nlm.nih.gov/pubmed/25211013>

Reviewer #2 (Remarks to the Author):

Ebrahimi, Richards, et al. present a fascinating study of the determinants of alternative splicing of APOBEC3H, the function impact on 3H protein, and the effect on HIV restriction. Using existing genomics data sets, they identify a candidate 3bp deletion strongly associated with splicing of the A3H transcript, and go on to test this using a mini gene construct. The authors also show anti HIV effects as well as protease cleavage of the SV200 A3H protein, adding intriguing data to evolution and antiviral activities of this locus. This represents an important contribution to the study of APOBEC3 locus and its role in HIV restriction, and is likely to be of broad interest to the field.

The suggestion of human L1 exonization only occurring after the 3-bp deletion is also intriguing.

Minor:

Pg 4: sub-Saharan Africa 58%-- , maybe rounding but I get $.58^2 + 2*.58*.42 = 82\%$ of this population has at least 1 allele (assuming Hardy Weinberg Equilibrium)

Pg 12: potential interbreeding. I am not sure about this speculation about the potential origin of the 3bp deletion due to archaic introgression. This variant simply being an old allele that predates modern human/archaic split seems possible. Which of the A3H haplotypes do the archaic humans have? Also, it seems possible that the indicated sequence can recurrently mutate to lose the 3bp – perhaps even in non-human primates. A search of available primate whole genome data may offer evidence in support of this, but such an analysis should not be required for the current manuscript.

Reviewer #3 (Remarks to the Author):

The manuscript entitled "Genetic and mechanistic basis for APOBEC3H (A3H) alternative splicing, differential retrovirus restriction, and counteraction by HIV-1 protease" by Ebrahimi et al. investigated the mechanism of A3H alternative splicing in haplotypes I and II as well as the consequences of the different splice variants on HIV-1 infectivity in a Vif-deficient context. Using bioinformatics analyses of RNAseq data the authors first discovered a link between the A3H haplotype II and the presence of the splice variant SV200. They have shown that the SV200 transcript as well as the associated protein is significantly enriched in haplotype II homozygous individuals compared to haplotype I and this is strongly linked to the presence of a trinucleotide deletion located in intron 4 of the haplotype II gene. In the absence of Vif, the three studied splice variants, SV182, SV183 and SV200 are encapsidated into virus-like particles and considerably decrease their infectivity in a single-cycle infection. The authors showed that the SV200 splice variant is cleaved inside viral particles and this is likely due to the HIV-1 protease. Point mutations have been identified that enhance or inhibit this cleavage and their effect on HIV-1 restriction has been investigated. Ebrahimi and colleagues finally propose that A3H cleavage by the viral protease confers another layer of protection in addition to Vif-mediated counteraction of A3H restriction.

The manuscript of Ebrahimi and coworkers is addressing a very interesting question on the most complex APOBEC3 protein. Indeed, A3H not only has many different haplotypes but also different splice variants which considerably complicates the study of this restriction factor. I appreciate the authors efforts to elucidate this issue. This manuscript is well written, clearly structured and contributes considerably to the understanding of the two main A3H haplotypes by pinpointing their main differences, which seems to be due to splicing.

In my opinion, there are only a few minor issues to address.

Here are my comments and suggestions that I hope may help Ebrahimi and co-workers at improving clarity and comprehension of their findings:

- Figure 2: The percentage of transcripts corresponding to the different splice variants of Hap II/II (fig. 2c) seems to be very homogenous. However, protein level (Fig. 2d) are higher for both SV182 and SV183 than for SV200. Could you comment this discrepancy? Do you think these transcripts (sequence, structure) could potentially influence their translation?
- Figure 4: The minigenome used in this experiment has the sequence of haplotype II and only the ctc trinucleotide is inserted. It would be interesting to do the reverse experiment by using the background of the haplotype I and deleting the ctc trinucleotide. If you compare the level of A3H SV200 protein (panel b-left) to the one in figure 2d, it seems that the ratio SV200/SV182-3 is much lower. Could you please comment? Do you think the deletion of introns 1-3 could have influenced the protein level of A3H SV200?
- Figure 5a: to what extent do you think SV200 is physiologically relevant in infection, given that it seems expressed and encapsidated to a much lesser extent than SV182 and SV183?
- Figure 5: experiments have been performed in a Vif-null background. What is the effect of Vif on these different A3H splice variants?

- Figure 6a: this panel is not intuitive to me because at first, it seems that the four A3H forms (SV200, V185Q, R186A and SV183) possess the same restrictive activity, which is clearly not the case. Please provide a clearer explanation in the main text and figure legend. Does statistical analysis show a significant difference between the percentage of restriction of the three forms?
- The figure legends of figure 2a and b are swapped.
- The RNAseq profiles in figure 2b show some colored blocks. Could you explain the meaning of these colors?

Detailed Responses to Reviewer Comments

Reviewer #1:

“Ebrahimi et al show that the most antiviral of the APOBEC3H haplotypes, haplotype II, also encodes a splice variant that creates a new exon. This splice variant makes a larger protein with increased antiviral activity, and is cleaved by HIV protease. While the data looks sound, there is some hyperbole related to its interpretation.”

Response: We thank this reviewer for appreciating the soundness of our data sets.

“The major problem of the interpretation that the APOBEC3H SV200 splice variant has important antiviral activity, is that there is very little of it made relative to the other splice variants. Figure 2D is a good example of this where there looks like there is much less of the large band relative to the other bands that react with the antibody. Given how APOBEC3 proteins have dose-dependent effects, I am not certain that this splice variant really has that much of a role.”

Response: We very much appreciate this comment, and we now address it by quantifying the bands representing SV200 and SV182/3 (indistinguishable) in immunoblots in **Fig. 2** and additional biologically independent experiments not shown. Densitometry values were used to calculate the percentage of SV200 among total detectable A3H protein (SV200 and SV182/3) in homozygous haplotype II PBMCs and LCLs. Quantification showed that the SV200 band comprises 30-35% of the total A3H protein in PBMCs (n = 4; mean = 32 +/- 1.9 SD) and 16-43% in LCLs (n = 8; mean = 24 +/- 9.3). The PBMC data are entirely concordant with splice variant mRNA quantification in **Fig. 2c** where each variant is present in equal amounts (ca. 33% each). The LCL values are more variable but still within 2-fold within 2-fold of the primary cell data. The primary cell (PBMC) values are more likely to be representative of the *in vivo* protein levels, thus, we infer that this major splice variant has antiviral roles that are at least equal to those of SV182 and SV183 in cells.

The text has been revised as follows (page 7): “Importantly, quantification of the amount of SV200 in immunoblots from homozygous haplotype II PBMCs and LCLs showed that this variant comprises 30-35% and 16-43%, respectively, of total A3H protein (n=4 and n=8 independent experiments, respectively; **Fig. 2d-e** and additional blots not shown). This result is concordant with SV200 constituting one-third of total A3H haplotype II splice products (**Fig. 2c**) and approximately one-third of total cellular A3H protein (**Fig. 2d-e**). Together, these data demonstrate that the SV200 enzyme is a major expressed splice variant with similar opportunities (based on abundance) to contribute to antiviral immunity as SV182 and SV183.”

“Figure 2E is confusing to me because the APOBEC3H bands increase in intensity after HIV infection. Shouldn't the Vif protein degrade the intracellular pools of APOBEC3H? In any case, showing if Vif has a differential effect on SV200 (i.e. that perhaps SV200 is more resistant to Vif) would add to the interest in this splice variant.”

Response: We apologize for the confusion. In this experiment Vif-deficient HIV-1 (stop codons at Vif residues 26 and 27) was used as a tool to induce A3H expression in CD4+ primary T cells and increase signal for immunoblot experiments (concordant with published work, Fig. 1 in PMID 21835787). We now clarify this approach by modifying the panel label to read “HIV-1

Vif-XX”, and adding this important information to the figure legend and **Methods** section (subsection titled “**Endogenous A3H Immunoblots**”).

“The authors hypothesize that the 3bp deletion that gives rise to the splice variant is human-specific. However, I did not see where they compared the intron sequences to chimpanzees to determine if this is true or not. This impacts much of their discussion about the adaptation of HIV to humans.”

Response: The phylogenetic analysis in **Fig. 3c** shows that Δ ctc is specific to the human haplotype II genotype. In addition, we now include genomic and transcriptomic data from chimpanzee and rhesus macaque, which further indicates that 3bp deletion and SV200 expression are specific to the human haplotype II genotype (new **Supplementary Fig. 4**).

“Similar to comment 3, the authors hypothesize that HIV-1 protease adapted to counter-act the SV200 variant. However, they do not test this hypothesis by determining if other related lentiviruses, e.g. SIV_{mac} or better SIV_{cpz}, fail to cleave SV200 when this variant is co-transfected with the appropriate proviruses. A null hypothesis is that adaptation of protease did not occur, but just by chance the SV200 variant has a structure that just happens to be similar to one recognized by protease.”

Response: We thank this reviewer for making this important point. We now include new VLP immunoblot data in **Fig. 5d** showing that HIV-1 isolates representing sub-type B (transmitted founder and chronic circulating) and sub-type C (transmitted founders) are all capable of proteolytic cleavage of A3H-hapII SV200 (cleavage is incomplete and variable most likely due to different amounts of mature vs immature VLPs). In contrast, SV200 is not cleaved by the protease of SIV_{mac239}. We tried several SIV_{cpz} molecular clones from Beatrice Hahn, but these are unstable and picked up deletions during proviral plasmid preparation in *E. coli* resulting in no production of virus. Nevertheless, these additional results indicate that proteolytic cleavage of A3H is likely to be conserved for HIV-1. We suggest in **Discussion** that HIV-1 had to adapt to the increased restriction capacity of A3H haplotype II (SV182, SV183, and SV200) because the Δ ctc SNP that enables SV200 production is far more ancient than the entry of HIV-1 into the human population (1000’s versus 100’s of years, respectively).

“Try as I might, I just do not understand Figure 6. For one thing, there seems to be no dose-response in the expression of most of the APOBEC3H proteins. This would seem to indicate that they are already at saturation at the lowest doses, in which case, the calculations of abundance are not meaningful. Also, the Y axis of percent restriction is such a narrow range (40 to 80, so only a 2-fold range). I think this experiment needs to be done within a large linear range, and again, as in point #1, I am concerned that the levels of the splice variant relative to the major variant are too low to be biologically important.”

Response: We have rewritten the text to be as clear as possible but we are not sure what else can be done. We already show raw data from 3 biologically independent experiments in **Fig. 6b**, with means +/- SEMs plotted in **Fig. 6a**. A broader titration has not been done because lower protein amounts would be undetectable and higher amounts are likely to result in complete restriction and/or saturate immunoblot signals.

“Page 3, top: Yes, bats with 13 genes is a lot, but is not "incredible." Humans have 14 interferon alpha genes, for example.”

Response: We have revised this part of the text to read “... For instance, gene copy numbers vary from 1 gene in rodents (encoding 2 deaminase domains)¹³, 7 genes in most humans (11 deaminase domains)^{13,14}, and up to 13 genes in bats (18 deaminase domains)¹⁵.”

“Page 3, top: No, the number of SNPs and insertions/deletions in APOBEC3 genes is not especially profound compared to many other genes. Also, it is important not to confuse this with positive selection as the text seems to do (ie. one is within a species, the other is between species).”

Response: We have revised this part of the text to read “... Additionally, there are coding and non-coding single nucleotide polymorphisms (SNPs) and insertion/deletion polymorphisms (indels), which cumulatively account for high rates of positive selection (reviewed by refs.¹⁸⁻²⁰).”

Page 3, middle: "...there is a continuous battle.." is a little over-dramatic for a scientific paper.”

Response: We have revised this part of the text to read “... Thus, during every round of virus replication, there is a continuous interaction between A3 enzymes restricting virus and Vif neutralizing A3s (reviewed by refs.^{10,11,21}).”

Page 11, bottom: No, this is not the first description of a coding region formed from an L1 element. See <https://www.ncbi.nlm.nih.gov/pubmed/25211013>”

Response: We apologize for not phrasing this correctly. We have now revised this part of the text to read “...to our knowledge, this is the first example of L1 exonization of a primate A3 gene. This is ironic given that A3H has been shown to elicit strong L1 restriction activity^{17,67,68}.”

Reviewer #2:

“Ebrahimi, Richards, et al. present a fascinating study of the determinants of alternative splicing of APOBEC3H, the function impact on 3H protein, and the effect on HIV restriction. Using existing genomics data sets, they identify a candidate 3bp deletion strongly associated with splicing of the A3H transcript, and go on to test this using a mini gene construct. The authors also show anti HIV effects as well as protease cleavage of the SV200 A3H protein, adding intriguing data to evolution and antiviral activities of this locus. This represents an important contribution to the study of APOBEC3 locus and its role in HIV restriction, and is likely to be of broad interest to the field.”

Response: We thank this reviewer for acknowledging the importance and potential broad interest of our studies.

“Pg 4: sub-Saharan Africa 58%-- , maybe rounding but I get $.58^2 + 2*.58*.42 = 82\%$ of this population has at least 1 allele (assuming Hardy Weinberg Equilibrium)”

Response: We thank this reviewer for pointing this out. Based on the 1000 Genomes database, the allele frequency of A3H hapII in sub-Saharan Africa is 56.7%, which means 81.5% of this population has at least one allele. In the revised text, we use rounded frequencies (page 4): “... although the global frequency of the *A3H* haplotype II is 20.9%, the allele frequency in HIV-1 pandemic areas of sub-Saharan Africa is 57%, which means 82% of this population has at least one allele of this HIV-1 restrictive haplotype.”

Pg 12: potential interbreeding. I am not sure about this speculation about the potential origin of the 3bp deletion due to archaic introgression. This variant simply being an old allele that predates modern human/archaic split seems possible. Which of the A3H haplotypes do the archaic humans have? Also, it seems possible that the indicated sequence can recurrently mutate to lose the 3bp – perhaps even in non-human primates. A search of available primate whole genome data may offer evidence in support of this, but such an analysis should not be required for the current manuscript.”

Response: We fully appreciate this point. To investigate the source of Δ ctc in modern hap II humans, we first examined short stretches of sequences immediately flanking Δ ctc. We observed that these regions are conserved in modern and archaic humans, and even in chimpanzee (new **Supplementary Fig. 4**). In other words there were not additional SNPs to enable us to determine whether recombination has been the source of Δ ctc or it was caused by an archaic-independent deletion event, possibly mediated by DNA polymerases, which are known sources of indels, particularly in repetitive sequence tracts⁶⁹. To further investigate the source of Δ ctc we next inspected the entire ~7 kb sequence of Neanderthal and Denisovan A3H genes and found an additional 13 SNPs, which were specific to A3H hapII and both archaic genomes and were not present in the genome of other modern humans. In contrast, there was only one SNP that was common to A3H hap I and archaic genomes, and it was different in human hapII. To confirm these observations, we performed a phylogenetic analysis, which showed that human *A3H* hapII gene strongly groups with Neanderthal and Denisovan *A3H* genes. Nevertheless, we do not present additional data here, because we prefer to address this question separately and in collaboration with experts in the evolutionary biology field.

Accordingly, we have revised the relevant part of our discussion to read (page 13): “...The original Δ ctc may have independently occurred one or more times in an ancestral *A3H*

haplotype II population, as it does not exist (apart from rare recombinants) in other modern human *A3H* haplotype populations. An alternative hypothesis is that Δ ctc emerged in modern hapII humans as a result of recombination with an archaic genome. An analogous Δ ctc variant exists in Neanderthal and Denisovan genomic DNA sequences (**Supplementary Fig. 4**). Moreover, within the 7 kb *A3H* gene there are an additional 13 SNPs specific to archaic and modern humans with the haplotype II genotype. In contrast only one SNP is shared between archaic and modern humans with the haplotype I genotype. Altogether, these data suggest that Δ ctc may have an archaic origin. However, further studies are needed to pinpoint the exact source of this important variant.”

Reviewer #3:

“The manuscript entitled “Genetic and mechanistic basis for APOBEC3H (A3H) alternative splicing, differential retrovirus restriction, and counteraction by HIV-1 protease” by Ebrahimi et al. investigated the mechanism of A3H alternative splicing in haplotypes I and II as well as the consequences of the different splice variants on HIV-1 infectivity in a Vif-deficient context. Using bioinformatics analyses of RNAseq data the authors first discovered a link between the A3H haplotype II and the presence of the splice variant SV200. They have shown that the SV200 transcript as well as the associated protein is significantly enriched in haplotype II homozygous individuals compared to haplotype I and this is strongly linked to the presence of a trinucleotide deletion located in intron 4 of the haplotype II gene. In the absence of Vif, the three studied splice variants, SV182, SV183 and SV200 are encapsidated into virus-like particles and considerably decrease their infectivity in a single-cycle infection. The authors showed that the SV200 splice variant is cleaved inside viral particles and this is likely due to the HIV-1 protease. Point mutations have been identified that enhance or inhibit this cleavage and their effect on HIV-1 restriction has been investigated. Ebrahimi and colleagues finally propose that A3H cleavage by the viral protease confers another layer of protection in addition to Vif-mediated counteraction of A3H restriction.

The manuscript of Ebrahimi and coworkers is addressing a very interesting question on the most complex APOBEC3 protein. Indeed, A3H not only has many different haplotypes but also different splice variants which considerably complicates the study of this restriction factor. I appreciate the authors efforts to elucidate this issue. This manuscript is well written, clearly structured and contributes considerably to the understanding of the two main A3H haplotypes by pinpointing their main differences, which seems to be due to splicing.”

Response: Thank you for appreciating our results.

“Figure 2: The percentage of transcripts corresponding to the different splice variants of Hap II/II (fig. 2c) seems to be very homogenous. However, protein level (Fig. 2d) are higher for both SV182 and SV183 than for SV200. Could you comment this discrepancy? Do you think these transcripts (sequence, structure) could potentially influence their translation?”

Response: We previously quantified splice variant transcript levels (**Fig. 2c**), and we now include protein level quantifications. Densitometry values were used to calculate the percentage of SV200 among total detectable A3H protein (SV200 and SV182/3) in homozygous haplotype II PBMCs and LCLs. Quantification showed that the SV200 band comprises 30-35% of the total A3H protein in PBMCs (n = 4; mean = 32 +/- 1.9 SD) and 16-43% in LCLs (n = 8; mean = 24 +/- 9.3). The PBMC data are entirely concordant with splice variant mRNA quantification in **Fig. 2c** where each variant is present in equal amounts (ca. 33% each). The LCL values are more variable but still within 2-fold within 2-fold of the primary cell data. The primary cell (PBMC) values are more likely to be representative of the *in vivo* protein levels, thus, we infer that this major splice variant has antiviral roles that are at least equal to those of SV182 and SV183 in cells.

The text has been revised as follows (page 7): “Importantly, quantification of the amount of SV200 in immunoblots from homozygous haplotype II PBMCs and LCLs showed that this variant comprises 30-35% and 16-43%, respectively, of total A3H protein (n=4 and n=8 independent experiments, respectively; **Fig. 2d-e** and additional blots not shown). This result is concordant with SV200 constituting one-third of total A3H haplotype II splice products (**Fig. 2c**) and approximately one-third of total cellular A3H protein (**Fig. 2d-e**). Together, these data

demonstrate that the SV200 enzyme is a major expressed splice variant with similar opportunities (based on abundance) to contribute to antiviral immunity as SV182 and SV183.”

“Figure 4: The minigenome used in this experiment has the sequence of haplotype II and only the ctc trinucleotide is inserted. It would be interesting to do the reverse experiment by using the background of the haplotype I and deleting the ctc trinucleotide. If you compare the level of A3H SV200 protein (panel b-left) to the one in figure 2d, it seems that the ratio SV200/SV182-3 is much lower. Could you please comment? Do you think the deletion of introns 1-3 could have influenced the protein level of A3H SV200?”

Response: We have not done the reverse experiment because the haplotype I protein is poorly expressed and we predict that it will be very hard to perform the necessary immunoblot experiments to confirm alternative splicing at the protein level.

“Figure 5a: to what extent do you think SV200 is physiologically relevant in infection, given that it seems expressed and encapsidated to a much lesser extent than SV182 and SV183?”

Response: Please see the point above on quantification. SV200 is expressed similarly to SV182 or SV183 at both mRNA and protein levels in PBMCs from homozygous A3H haplotype II individuals. Therefore, it should be considered a major splice variant capable of exerting physiologically relevant antiviral activities.

“Figure 5: experiments have been performed in a Vif-null background. What is the effect of Vif on these different A3H splice variants?”

Response: This is a valid question that is already addressed in part by prior studies from the Simon lab, who originally reported these splice variants (PMID 18945781). They showed that NL4-3 Vif, which is the same as IIB Vif, is incapable of counteracting FLAG-tagged A3H SV183 or SV200 (Fig. 4 in PMID 18945781). However, we and others including the Simon lab subsequently showed that these Vif variants are hypofunctional against A3H, despite having fully degradation activity against A3G (e.g., PMID 25411794). Hyperfunctional Vif variants such as LAI Vif and some natural isolate Vifs have stronger A3H neutralization/degradation activities. These differences are further complicated by epitope tag identity, tag placement, and whether Vif is expressed from a simple overexpression vector or in the context of a full-length molecular clone. In short, the results of this suggested experiment will depend heavily on the Vif protein that is used and the overall experimental system. Thus, we would prefer to address it properly through dedicated future studies pitting different Vifs against the different splice variants in multiple experimental systems (single cycle and spreading infection in cell lines and primary lymphocytes with key A3H genotypes). We also feel that the results of these studies will be tangential to our current story focusing on the A3H haplotype II splice variant SV200, the enhanced activity of the resulting SV200 protein, and the novel HIV-1 protease-dependent counteraction mechanism that blunts the activity of this protein.

“Figure 6a: this panel is not intuitive to me because at first, it seems that the four A3H forms (SV200, V185Q, R186A and SV183) possess the same restrictive activity, which is clearly not the case. Please provide a clearer explanation in the main text and figure legend. Does statistical analysis show a significant difference between the percentage of restriction of the three forms?”

Response: We have revised the text and legend as recommended. The Y-axis in **Fig. 6a** is percent restriction and, indeed, all enzymes achieve maximal restriction of Vif-deficient HIV-1. However, as shown by the X-axis, the SV200 enzyme achieves this at much lower protein concentrations. The raw data from 3 biologically independent experiments are shown **Fig. 6b** with means +/- SEMs plotted in **Fig. 6a**. Apart from showing these confidence intervals we are not sure how to perform additional statistical analyses for this large data set because neither the restriction levels nor the A3H encapsidation levels perfectly match-up for any condition (*i.e.*, neither X nor Y values are fixed despite best attempts to transfect similar DNA amounts for each titration series).

“The figure legends of figure 2a and b are swapped.”

Response: Thank you for catching this. The mistake has been fixed.

“The RNAseq profiles in figure 2b show some colored blocks. Could you explain the meaning of these colors?”

Response: The colored blocks indicate polymorphic positions based on the default presentation in the Integrative Genomic Viewer program from the Broad Institute (<https://software.broadinstitute.org/software/igv/>).

REVIEWERS' COMMENTS:

Reviewer #1 (Remarks to the Author):

The authors have addressed my concerns.

Reviewer #2 (Remarks to the Author):

I am satisfied with the author's response. The indicated changes have improved and clarified the manuscript.

Reviewer #3 (Remarks to the Author):

The paper has been greatly improved and fulfill the comments raised during the first round of reviewing.